# Optimized Wavelength Sampling for Thermal Radiative Transfer in Numerical Weather Prediction Models

**Michael de Mourgues [1], Claudia Emde [1,2] and Bernhard Mayer [1,2,*]**

1 Meteorologisches Institut, Ludwig-Maximilians-Universität (LMU), 80333 Munich, Germany
2 Deutsches Zentrum für Luft- und Raumfahrt, Institut für Physik der Atmosphäre, Oberpfaffenhofen, 82234 Weßling, Germany
* Correspondence: bernhard.mayer@physik.uni-muenchen.de

**Abstract:** In the thermal spectral range, there are millions of individual absorption lines of water vapor, $CO_2$, and other trace gases. Radiative transfer calculations of wavelength-integrated quantities, such as irradiance and heating rate, are computationally expensive, requiring a high spectral resolution for accurate numerical weather prediction and climate modeling. This paper introduces a method that could highly reduce the cost of integration in the thermal spectrum by employing an optimized wavelength sampling method. Absorption optical thicknesses for various trace gases were calculated from the HITRAN 2012 spectroscopic dataset using the ARTS line-by-line model as input to a fast Schwarzschild radiative transfer model. Using a simulated annealing algorithm, different optimized sets of wavelengths and corresponding weights were identified, which allowed for accurate integrated quantities to be computed as a weighted sum, reducing the computational time by several orders of magnitude. For each set of wavelengths, a lookup table, including the corresponding weights and absorption cross-sections, is created and can be applied to any atmospheric setups for which it was trained. We applied the lookup table to calculate irradiances and heating rates for a large set of atmospheric profiles from the ECMWF 91-level short-range forecast. Ten wavelength nodes are sufficient to obtain irradiances within an average root mean square error (RMSE) of upward and downward radiation at any height below $1\,\mathrm{Wm}^{-2}$, while 100 wavelengths allowed for an RSME of below $0.05\,\mathrm{Wm}^{-2}$. The applicability of this method was confirmed for irradiances and heating rates in clear conditions and for an exemplary cloud at 3.2 km height. Representative spectral gridpoints for integrated quantities in the thermal spectrum (REPINT) is available as absorption parameterization in the libRadtran radiative transfer package, where it can be used as an efficient molecular absorption parameterization for a variety of radiative transfer solvers.

**Keywords:** simulated annealing; radiative transfer; numerical weather prediction; thermal infrared; gas absorption; earth's atmosphere; heating rates

## 1. Introduction

Quantities such as irradiance and heating rates have complex spectral line structures, which vary widely from one atmospheric scenario to another. The complexity of spectral irradiance is demonstrated in Figure 1. The values of spectrally integrated irradiances and heating rates depend on the vertical prevalence and distribution of different trace gases in the atmosphere. Water vapor plays an important role, but variations in other trace gas concentrations, as well as clouds, have a large impact on their integrated value. Figure 2 shows simulated irradiances at the top of the atmosphere, without gas absorption, and irradiances including absorption by only one of the trace gases $H_2O$, $CO_2$, and $O_3$ at a time.

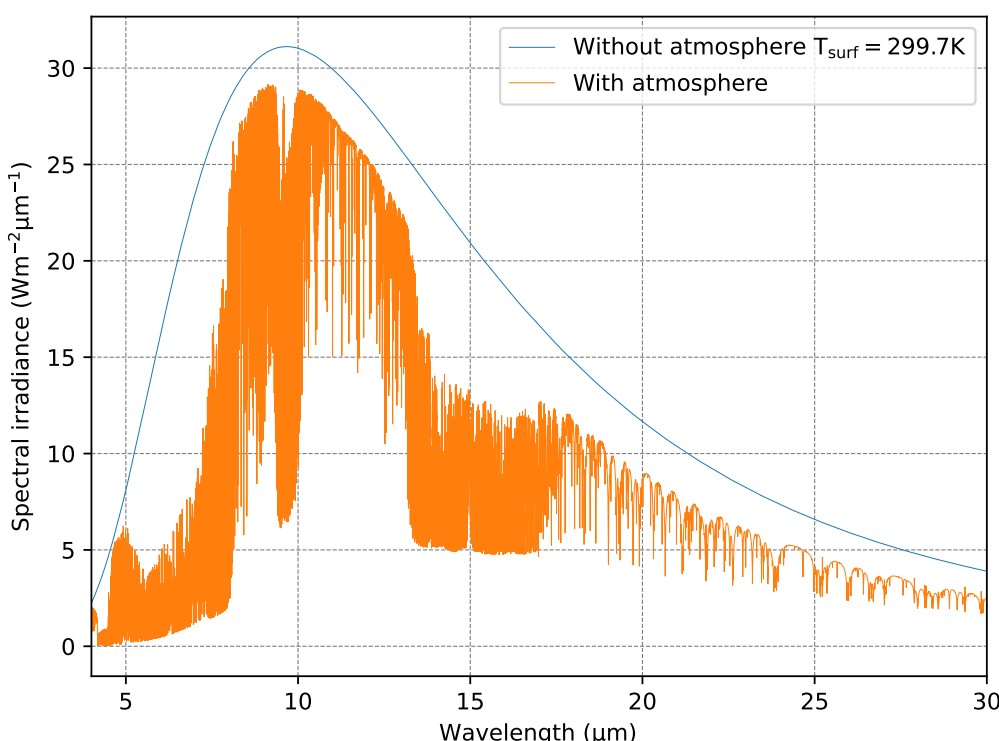

**Figure 1.** Spectral upward irradiance at the ground and at the top of the atmosphere, considering emission and absorption for an exemplary atmosphere in the dataset of [1].

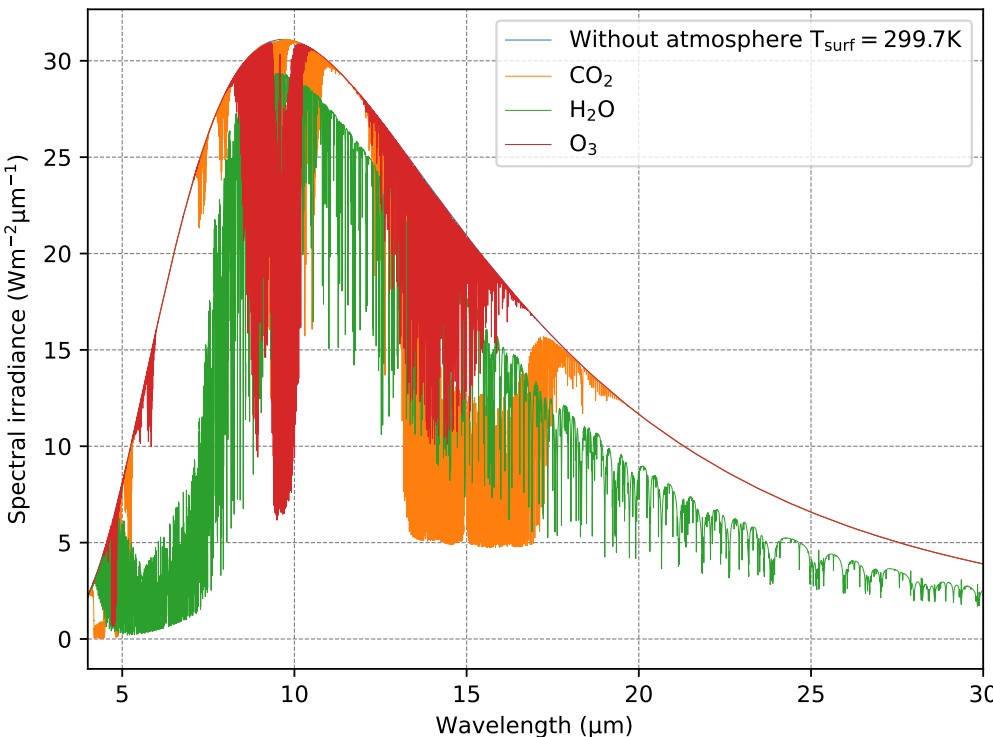

**Figure 2.** Spectral upward irradiance at the ground and at the top of the atmosphere, considering the emission and absorption of different trace gases separately for the atmosphere from Figure 1.

To model the irradiances throughout the atmosphere, one needs to represent the complex structure of the spectral absorption. A straightforward approach to calculating

spectrally integrated irradiances would be to choose a fine spectral resolution with a high number of sampling nodes. This has the disadvantage that about 100,000 nodes are required for a precise calculation of the thermal spectrum, which is computationally expensive. To minimize computational time, smarter methods have to be applied.

One such method [2] is the correlated-k approximation approach. Instead of integrating radiative quantities over wavelength nodes, this method introduces a distribution function of absorption cross-sections, which defines a corresponding probability for their occurrence. This distribution is a more tractable, continuous function, in contrast to a spectrum. To calculate the irradiance, an integral over the smoother distribution of the absorption cross-section space is performed. This approach is also recurring in a recent paper by [3], where user-specified bands are partitioned into subbands, and the correlated-k approximation is applied. Several groups are working on such parameterizations at present, employing low amounts of wavelength bands to reduce computational cost.

The methods introduced by [4,5] for shortwave resp. longwave radiative transfer avoid monochromatic calculations by considering equivalent grey systems instead. The thermal spectrum is then separated into spectral bands, which are each parameterized via the Malkmus band-model.

A different approach is proposed by [6,7], who introduced methods using a weighted mean with representative wavelengths. These methods seek to identify wavelength sampling nodes, for which the monochromatic atmospheric quantities determine the optimal corresponding integrated quantities for different atmospheric scenarios using a weighted mean. Both approaches include a training set, on which well-performing wavelength nodes are identified and subsequently verified by a different testing set. In [6], the targets of the method were radiances for satellite channels, while in [7], the radiances and irradiances for spectral bands of different widths, as well as satellite channels, were targeted.

This work uses a similar approach, but instead of targets, the integrated thermal spectrum is used for optimization. The introduced method allows for the calculation of integrated irradiances and heating rates for any atmospheric scenario while reducing the computational time by several orders of magnitude compared to line-by-line calculations. An optimal set of sampling nodes, as well as corresponding weights, are determined to replace the integration over wavelengths by a weighted sum. The method involves the following steps: first, from a training set of atmospheres [1], the corresponding altitude-dependent absorption optical thicknesses are calculated. These are obtained through the atmospheric radiative transfer simulator (ARTS) line-by-line model by [8] using the high-resolution transmission molecular absorption database (HITRAN) 2012 line catalogue [9] as input. We applied the lookup-table method described by [10], which allows for us to obtain absorption cross-sections for arbitrary temperatures and pressures by interpolation from a lookup-table. Additional interpolation in teh water-vapor mixing ratio is necessary due to the self-interaction between water-vapor molecules. The next step involved the calculation of spectral irradiances using the previously calulated optical thicknesses by solving the radiative transfer equation, neglecting scattering (Schwarzschild approximation) in the thermal spectral range. This approximation can used to find representative wavelengths, since scattering properties are a comparatively smooth wavelength function. Please note that this simplified solver is only used to determine an optimized set of wavelengths, which can then be applied, together with any radiative transfer solver of arbitrary complexity. The simplification is only required to keep the computational cost of the optimization process reasonably low. For each atmosphere layer in the training dataset, upward and downward high-spectral-resolution (HSR) irradiances, as well as their integral over the full thermal spectral range, were calculated. Using the simulated annealing algorithm, as described in [6], a set of wavelength nodes and corresponding weights were determined, so that the weighted sum of the monochromatic irradiances could optimally approximate the integrated irradiances in the training dataset.

The determined set of wavelength nodes and weights was then applied to a different testing dataset for verification. For this purpose, subsets of the 5000 atmospheres in the

dataset by [11] were employed. This dataset covers the complete spatial and seasonal variability in atmospheric profiles sampled to verify radiation models. We tested the applicability of the method for clear and cloudy atmospheres, as well as for different radiative transfer solvers.

## 2. Thermal Radiative Transfer Model

Since the scattering properties are smooth (compared to molecular absorption) wavelength functions, these suffice to model the radiative transfer without scattering in a plane-parallel setting for the optimization process. To simulate the irradiances and heating rates under different atmospheric scenarios, the Schwarzschild equation was solved

$$\mu \mathrm{d}L(\lambda) = -L(\lambda)\mathrm{d}\tau + B(\lambda, T)\mathrm{d}\tau \tag{1}$$

with the cosine of propagation zenith angle $\mu$, wavelength $\lambda$, radiance $L(\lambda)$, optical thickness $\tau$ and the Planck-function $B(\lambda, T)$.

Spectral upward and downward irradiance $E_{i+1}^{\mathrm{up}}, E_{i+1}^{\mathrm{dn}}$ on each level was recursively calculated via

$$E_{i+1}^{\mathrm{up/dn}}(\lambda) = E_i^{\mathrm{up/dn}}(\lambda)e^{-\frac{\tau_i(\lambda)}{\mu}} + \pi B(\lambda, T_i)(1 - e^{-\frac{\tau_i(\lambda)}{\mu}}) \tag{2}$$

where the index $i$ represents the level index for irradiance $E_i$ and the layer index for the absorption optical thickness $\tau_i$. Upward irradiance at the surface, as well as downward irradiance at the top of atmosphere, are set to

$$E_0^{\mathrm{up}} = \pi B(\lambda, T_{\mathrm{surf}}) \tag{3}$$

$$E_N^{\mathrm{dn}} = 0 \tag{4}$$

where $T_{\mathrm{surf}}$ denotes the surface temperature and $N$ the uppermost level of the atmospheric profile. Equation (2) is an approximate solution of the Schwarzschild Equation (1). To reduce computational cost, we considered only one propagation direction ($\mu = 0.5$) and used this as an approximation of the irradiance.

The heating rate $H$ is defined as the temperature tendency $\frac{dT}{dt}$, usually calculated from the divergence in the net flux. For the Schwarzschild model introduced above, this amounts to

$$H = \frac{g}{c_p} \frac{E_{\mathrm{up}}^{i+1} - E_{\mathrm{dn}}^{i+1} - E_{\mathrm{up}}^i + E_{\mathrm{dn}}^i}{p_i - p_{i+1}}. \tag{5}$$

where $p$ denotes the pressure, $g = 9.81 \ \mathrm{ms}^{-2}$ the standard acceleration of gravity and $c_p = 1004.67 \ \mathrm{Jkg}^{-1}\mathrm{K}^{-1}$ for the heat capacity of air.

Training and testing datasets contain vertical atmospheric profiles of the volume mixing ratio (VMR) of trace gases, pressures as well as temperatures. The VMR of a gas is defined as the volume fraction of the trace gas. Optical thicknesses were calculated from the pressure, temperature and given VMRs $\xi_{ij}$ on each layer for each trace gas in the following way

$$\tau_i(\lambda) = n_i \sum_j \xi_{ij} \sigma_{ij}(\lambda, \bar{p}_i, \bar{T}_i). \tag{6}$$

The $\sigma_{ij}$ are the absorption cross-sections of the considered trace gases, and $n_i$ is the number density of the air in layer $i$. For layer properties such as temperature, water vapor concentration and pressure, the average values on the adjacent levels were determined.

The number density $n_i$ of layer $i$ is determined using the hydrostatic equation:

$$n_i = (p_i - p_{i+1})\frac{N_A}{g M_d}, \tag{7}$$

$N_A$ being the Avogadro constant and $M_d = 0.0289 \ \mathrm{kg \ mol}^{-1}$ the molar mass of dry air.

Absorption cross-sections vary the most strongly with respect to wavelength, due to the discrete nature of the energy transitions allowed in the molecules contained in the atmosphere. For this reason, interpolation is not performed in the wavelength and absorption cross-sections are stored on a grid with 100,000 wavelength nodes.

The shape of the absorption lines also has a significant dependency on pressure and temperature due to Doppler- and pressure-broadening. In addition to the spectral lines, the continuum needs to be considered, which is a non-linear function of the VMR in case of the self-continuum. The absorption cross-section of water vapor, therefore, is dependent on its own VMR

$$\sigma_{i,\mathrm{H_2O}} = \sigma_{i,\mathrm{H_2O}}(\lambda, \bar{p}_i, \bar{T}_i, \bar{\xi}_{i,\mathrm{H_2O}}). \tag{8}$$

For the absorption line parameters, data from the HITRAN2012 molecular spectroscopic database were chosen [9], while MT_CKD (Mlawer-Tobin-Clough-Kneizys-Davies) version 1.0 was used as water vapor continuum model [12], which is consistent with HITRAN2012. With the atmospheric radiative transfer simulator ARTS [13], absorption cross-sections were calculated for 100,000 wavelengths between 4 µm and 200 µm on a set of different pressure, temperature and VMR values of $H_2O$ (called perturbations in ARTS) to encompass all atmospheric variations in the datasets of [1,11].

The parameters used in ARTS are specified in Table 1. The calculation of optical thicknesses from the lookup table was implemented on the basis of the method described in [10].

**Table 1.** Parameters used for the generation of a lookup table using ARTS. The non-linear species (NLS) perturbations enable a calculation of the absorption of cross-sections of $H_2O$ depending on their own concentration. Each perturbation corresponds to the factor by which the water vapor concentration of the reference atmosphere is altered.

| Parameter | Amount | Scope |
|---|---|---|
| T Perturbation | 9 | $(-120, 120)$ °C |
| NLS Perturbation | 5 | $(0, 10)$ |
| Pressure grid | 41 | $(110{,}000, 0.0006892)$ Pa |
| Gas species | 9 | $H_2O$, $CO_2$, $O_3$, $N_2O$, CO, |
| | | $CH_4$, $O_2$, $HNO_3$, $N_2$ |

As the absorption cross-sections vary with temperature, pressure, and in case of $H_2O$ with its own VMR, an interpolation method was developed, which optimally considers these dependencies. For a general trace gas, the method first identifies the position of the desired absorption cross-section $\sigma_{ij}(\bar{p}_i, \bar{T}_i)$ in the grid of the stored absorption cross-sections, producing the following four values:

$$\begin{array}{cc} \sigma(p_{\mathrm{low}}, T_{\mathrm{low}}) & \sigma(p_{\mathrm{up}}, T_{\mathrm{low}}) \\ \sigma(p_{\mathrm{low}}, T_{\mathrm{up}}) & \sigma(p_{\mathrm{up}}, T_{\mathrm{up}}) \end{array} \tag{9}$$

Next, a bilinear interpolation of the absorption of cross-sections in terms of pressure and temperature is conducted, via a polynomial of the form

$$P(\Delta p, \Delta T) = a\Delta p\Delta T + b\Delta p + c\Delta T + d \tag{10}$$

with

$$\Delta p = \bar{p}_i - p_{\mathrm{low}}, \quad \Delta T = \bar{T}_i - T_{\mathrm{low}} . \tag{11}$$

The polynomial parameters are determined via the four adjacent datapoints on the corners of its domain.

In the special case of water vapor, considering its dependence on its own VMR, the position of the desired absorption cross-section $\sigma_{ij}(\bar{p}_i, \bar{T}_i, \bar{\bar{\xi}}_{i,H_2O})$ in the grid of the stored absorption cross-sections produces eight values:

$$
\begin{array}{ll}
\sigma(p_{low}, T_{low}, \xi_{low,H_2O}) & \sigma(p_{up}, T_{low}, \xi_{low,H_2O}) \\
\sigma(p_{low}, T_{low}, \xi_{up,H_2O}) & \sigma(p_{up}, T_{low}, \xi_{up,H_2O}) \\
\sigma(p_{low}, T_{up}, \xi_{low,H_2O}) & \sigma(p_{up}, T_{up}, \xi_{low,H_2O}) \\
\sigma(p_{low}, T_{up}, \xi_{up,H_2O}) & \sigma(p_{up}, T_{up}, \xi_{up,H_2O}).
\end{array}
\tag{12}
$$

In this case, the interpolation via the polynomial in Equation (10) was first conducted in $\xi_{H_2O}$ and temperature for $p_{low}$ and $p_{up}$ each. Both values were then linearly interpolated with respect to the pressure $\bar{p}$.

The accuracy of the interpolation method was tested by comparison with line-by-line (LBL) calculations of optical thicknesses with ARTS in the U.S. Standard Atmosphere [14]. Upward and downward irradiance on each level were simulated using the optical thicknesses derived from the lookup table by interpolation, and optical thicknesses calculated line-by-line using ARTS. The absolute RMSE between the two simulations was $0.87\,\mathrm{Wm}^{-2}$. This discrepancy, however, did not affect the optimization process, since both the to be approximated values and the approximations use the same interpolation.

To include clouds in the simulations, the following simplified approach was implemented: for one layer of the atmospheric profile the absorption's optical thickness uniformly increased by $\tau_i = 5$ throughout the entire spectrum, a reasonable assumption since the cloud optical thickness only weakly depended on its wavelength.

## 3. Simulated Annealing Method

This section describes the chosen approach for finding optimal wavelength nodes using simulated annealing. This includes a closer description of the training set that was used dataset, as well as the implementation of the simulated annealing algorithm.

### 3.1. Training Dataset

For a training set, the 42 atmospheres from [1] were selected to cover a wide variety of atmospheres with respect to water vapor, ozone concentration and temperature. These were specified for 43 layers, providing information about each layer's pressure, temperature and height, and the volume mixing ratio (VMR) of the trace gases $H_2O$, $O_3$, $CO_2$, $N_2O$, $CO$, and $CH_4$. For applications to different climatic scenarios, these atmospheres were additionally modified in their concentrations of $CO_2$ and $CH_4$. For the data considered in this case, each atmosphere in the Garand set of atmospheres was considered for 0, 1 and 5 times the original concentration in $CO_2$ and $CH_4$. Hence, 378 atmospheric scenarios were defined as a training set, with a total of 32,508 integrated values for upward and downward irradiance at the respective atmospheric layer interfaces, as well as the corresponding spectral values for 100,000 wavelengths each.

### 3.2. Simulated Annealing Algorithm

For a dataset of 100,000 points, the number of possible combinations for even only 10 nodes exceeds the number of possible positions in a chess game and cannot be solved by brute force with a computer. We used the simulated annealing method [6] to determine the optimal choice of sampling nodes over short calculation times. Using this method for a fixed number of nodes $(\lambda_i)_{1 \le i \le n}$, their positions in the wavelength grid can be repeatedly changed at random and evaluated at each step to determine their eligibility for the representative wavelength method.

The first position of the sampling nodes is chosen at random. For each annealing iteration, the wavelength position of one of the sampling nodes is randomly chosen. A linear regression is then performed with respect to the exactly integral calculations and the

spectral values for the irradiances of all atmospheric scenarios in the training dataset. This produces spectral weights $w_j$, which provide the best approximation of the exact integrals

$$\int d\lambda E_i(\lambda) \approx \sum_{j=0}^{N} w_j E_i(\lambda_j) \, . \tag{13}$$

The resulting RSME is then compared to the RSME computed in the previous iteration. If it is lower, the new constellation of wavelengths is chosen. To reduce the chance of being trapped in a local minimum, a probability is still provided to be chosen if it is higher. This probability depends on the "annealing temperature"

$$S = S_{\text{start}} - \frac{n}{n_{\text{tot}}} (S_{\text{start}} - S_{\text{end}}) \tag{14}$$

where $n$ is the current annealing iteration and $n_{\text{tot}}$ the total number of iterations. The annealing temperature decreases with "cooling" compared to the chosen start annealing temperature.

A commonly used probability function [6] for a new RSME $\varepsilon_{\text{new}} > \varepsilon_{\text{old}}$ to be adopted is

$$P(\lambda_i) = e^{-\frac{1}{S}(\varepsilon_{\text{new}} - \varepsilon_{\text{old}})} . \tag{15}$$

We tested various functions and found that the following function provides the best results at the end of the annealing process:

$$P(\lambda_i) = e^{-\frac{1}{S}\frac{\varepsilon_{\text{new}}}{\varepsilon_{\text{old}}}} . \tag{16}$$

In Table 2, the general parameters used to create reduced lookup tables by simulated annealing are shown. The annealing start and end parameters determine how much fluctuation is allowed during the annealing process. A low annealing start would prevent the algorithm from trying further sampling node constellations, while a high annealing end does not give the algorithm the incentive to optimize error. The number of annealing datapoints is also an important tool to gauge precision compared to computational time. If the indicated amount falls short of the amount of irradiance calculated for the training dataset, a random subset of datapoints from the amount of the annealing datapoint number is chosen. By taking a subset of the atmospheric data in the training set, it is assumed that a set of sampling nodes that perform well on a random subset of the training set will also perform sufficiently well on the whole training set.

**Table 2.** Parameters used to create reduced lookup tables with simulated annealing.

| Annealing Parameters | Values |
|---|---|
| Total annealing iterations $n_{\text{tot}}$ | 10,000 |
| $S_{\text{start}}$ | 0.5 |
| $S_{\text{end}}$ | $1 \times 10^{-7}$ |
| Annealing datapoints | 15,000 |
| Sampling nodes | 10, 30, 50, 100 |

### 3.3. Absorption Parameterization for the Thermal Spectral Region

From the chosen sampling nodes, a *reduced lookup table* is created, containing all the necessary information from the original lookup table to calculate spectral optical thicknesses and irradiances. In this case, however, this is exclusively for the wavelengths that correspond to the chosen sampling nodes. This implies that, instead of a large dataset containing absorption cross-sections for 100,000 wavelengths, only a fraction of this information is stored. This reduces the calculation times of atmospheric quantities such as irradiance by several orders of magnitude.

Aside from the datapoints used in the interpolation process, the weights $w_j$ (see (Equation 13)) calculated by linear regression are contained in this table.

## 4. Performance of the Parameterization

This section examines how the annealing algorithm performs for different parameters and discusses the characteristics of the chosen sampling nodes. The amount of sampling nodes, as well as the amount of annealing iterations, contribute to a more accurate result.

In Figure 3, the performance of a reduced lookup table with 10 and 100 wavelength sampling nodes is displayed for different numbers of annealing iterations. For each datapoint, the average and standard deviation of thirty simulated annealing runs on a set of 6000 irradiance values are shown. It becomes clear that, for a given number of sampling nodes, the minimum error converges to a lower bound, which cannot be improved by the addition of more annealing iterations. The limit reached at the right end of the plot indicates the maximum possible accuracy for a given number of sampling nodes.

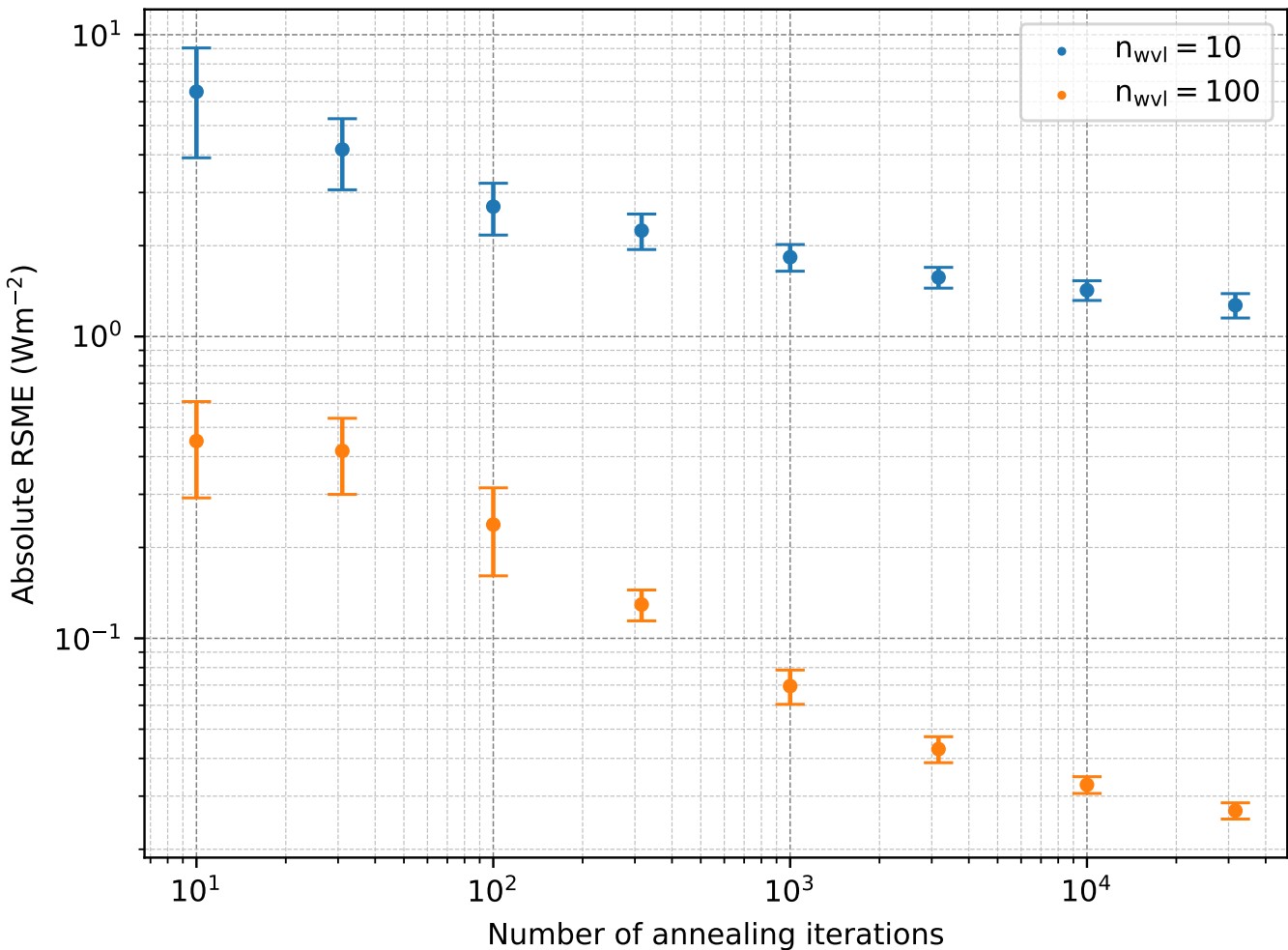

**Figure 3.** Performance of the simulated annealing algorithm on a set of 6000 irradiance datapoints of $E_{\text{up}}/E_{\text{up}}$ for different profiles at different heights. The average RSME of 30 runs for each number of annealing iterations is depicted on the graph for 10 resp. 100 sampling nodes.

As noted in [6], the values for annealing start temperature and end temperature, as well as the amount of annealing iterations, had to be chosen to avoid wasting computational time or depriving the algorithm of the chance to try new, potentially better wavelengths.

In Figure 3, the annealing result is shown to continually improve with the number of annealing iterations. It can be observed that there is significant potential for improving the

result, particularly for low iteration numbers. This progress is, however, slowing down so that further calculations would not improve the result and waste calculation time. For the purpose of these calculations, a total iteration number of $n_{tot} = 10,000$ was determined to be sufficient.

In Figure 4, the wavelengths and corresponding weights for four different annealing runs with different numbers of sampling nodes are displayed. Some information about the position and weights are summarized in Table 3. At least three conclusions can be drawn:

1. The linear regression method produces few negative weights. These did not produce any unphysical results during testing.
2. Sampling nodes concentrate on the part of the spectrum in which most of the absorption of different trace gases takes place. This can be seen by comparing to the impact of trace gas absorption ($CO_2$, $H_2O$ and $O_3$) in Figure 2.
3. Comparatively large weights at longer wavelengths, where little absorption takes place, account for the bulk of the integrated value, while sampling nodes at shorter wavelengths with high absorption determine the fine-tuned values with respect to specific atmospheric scenarios.

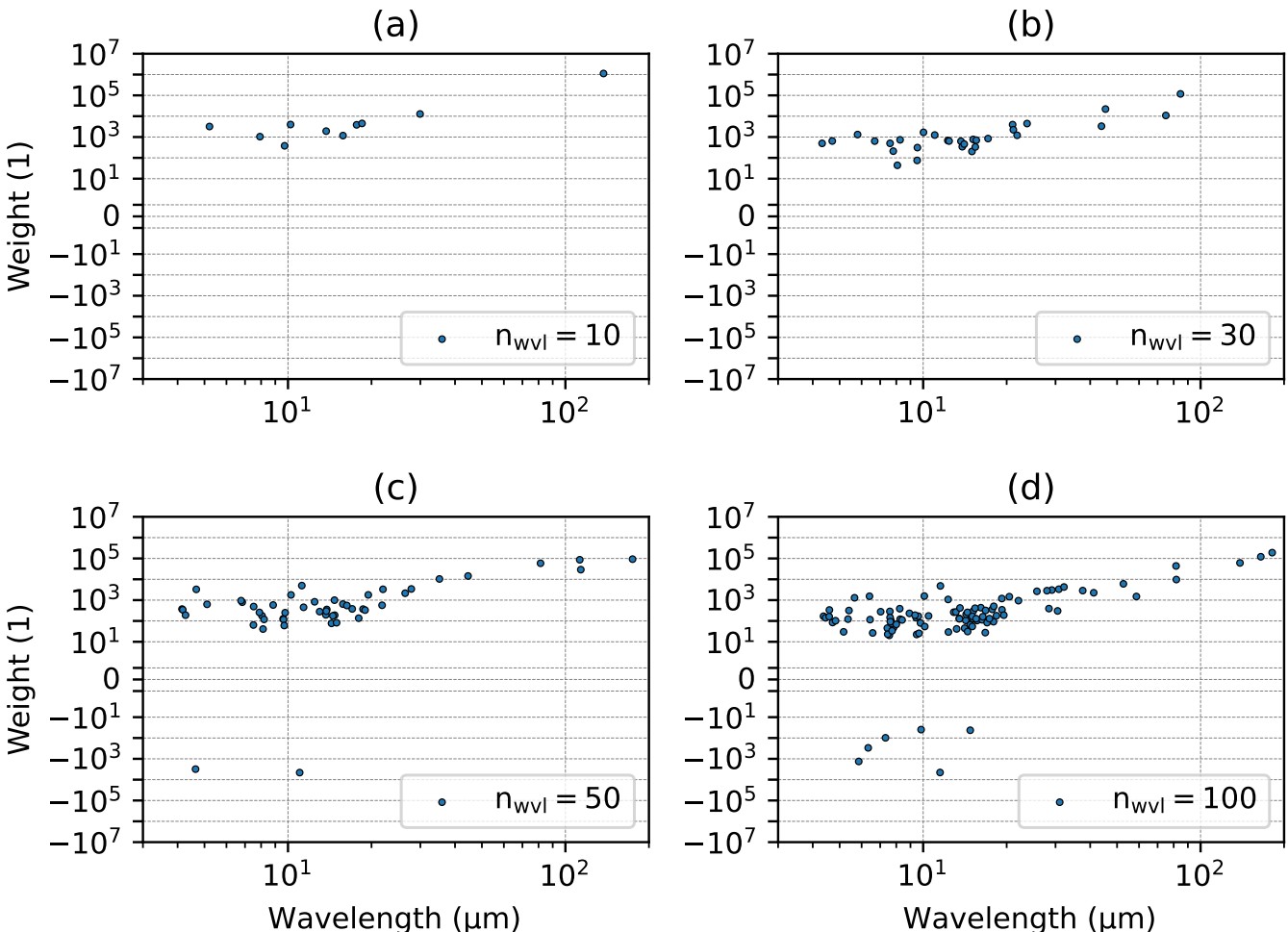

**Figure 4.** Positions of sampling nodes and chosen weights from reduced lookup tables for 10 (**a**), 30 (**b**), 50 (**c**) and 100 (**d**) sampling nodes.

**Table 3.** Fraction of sampling nodes in weight and wavelength ranges.

| $n_{wvl}$ | 4 μm $< \lambda <$ 20 μm | 4 μm $< \lambda <$ 40 μm | $|w| < 1000$ | $|w| < 5000$ | $w < 0$ |
|---|---|---|---|---|---|
| 10 | 0.70 | 1.00 | 0.20 | 0.70 | 0.00 |
| 30 | 0.73 | 0.87 | 0.63 | 0.90 | 0.00 |
| 50 | 0.80 | 0.90 | 0.70 | 0.88 | 0.04 |
| 100 | 0.82 | 0.92 | 0.77 | 0.94 | 0.06 |

### 4.1. Test Dataset

To test the method, we used a database of [11] containing 5000 profiles with 91 levels. These profiles were sampled from an even larger database containing 121,462,560 profiles from cycle 30R2 of the the European Centre for Medium-Range Weather Forecasts (ECMWF) forecasting system. The dataset provides an exhausting variation in atmospheric temperature and specific humidity. The database scenarios include clear and cloudy cases.

### 4.2. Accuracy of Irradiances

Figure 5 shows the absolute RSME of irradiances calculated by a weighted sum over representative wavelengths with respect to the accurate irradiances calculated by integration in the high-resolution spectrum. The blue dots correspond to lookup-tables including 10 representative wavelengths, the orange dots to 30 representative wavelengths, the green dots to 50, and the red dots to 100 representative wavelengths. For each reduced lookup table, the same randomly selected set of 500 atmospheres from the test dataset was used. As the figure shows, the training set of atmospheres by [1] is sufficient for the creation of reduced lookup tables, which perform equally well for all different atmospheric scenarios from [11].

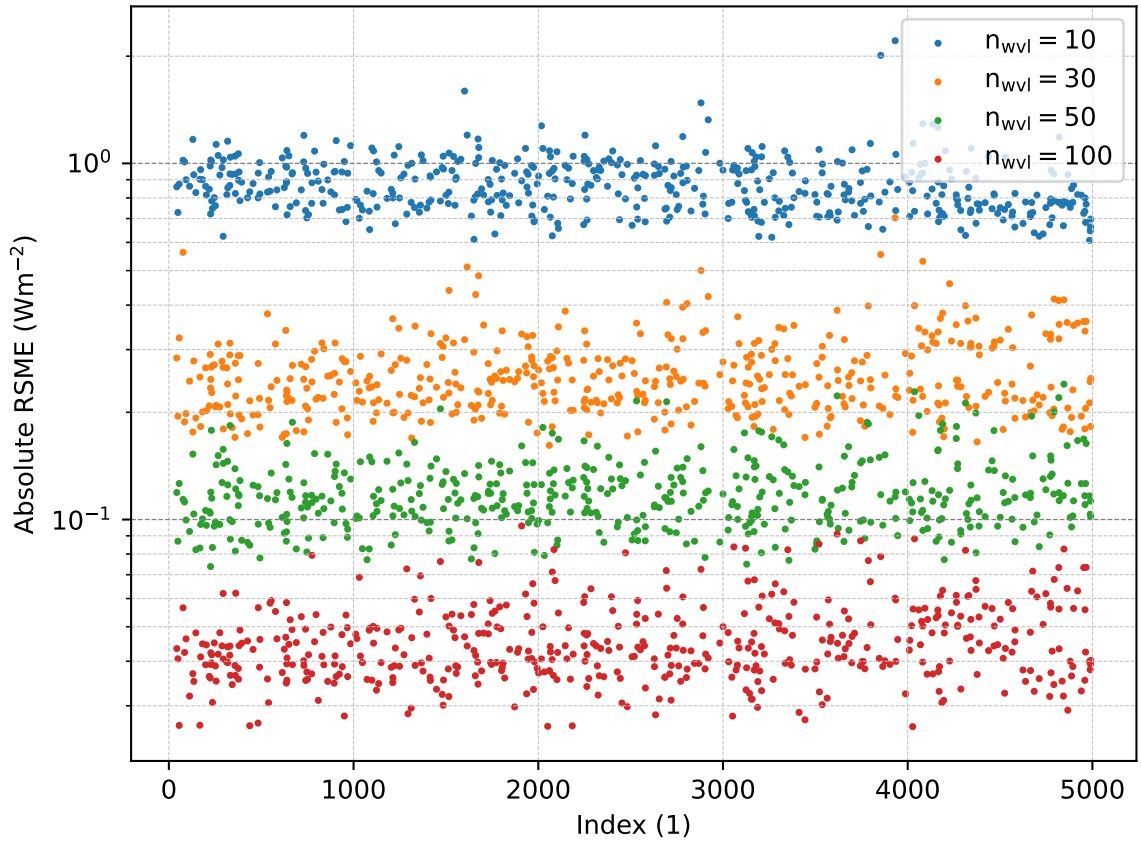

**Figure 5.** Absolute RSME of irradiance, as determined by weighted sum and high spectral resolution calculations, for 500 randomly chosen atmospheres in the dataset of [11].

The average errors and standard deviations of the 500 profiles in Figure 5 are shown in Table 4. As expected, a higher amount of sampling nodes improves the accuracy; however, the computational time linearly depends on the number of sampling nodes.

**Table 4.** Average $\mu$ and standard deviation $\sigma$ of the absolute RSME of radiative upward and downward irradiance $E_{up}$, $E_{dn}$ for 500 atmospheric dataset profiles by [11], for four different reduced lookup tables. Each calculation was made for the original atmosphere, as well as the same atmosphere with an added cloud. The unit for values in the table is [$Wm^{-2}$].

| | Clear | | Cloudy | |
|---|---|---|---|---|
| $n_{wvl}$ | $\mu$ | $\sigma$ | $\mu$ | $\sigma$ |
| 10 | 0.874 | 0.163 | 0.723 | 0.165 |
| 30 | 0.256 | 0.067 | 0.236 | 0.054 |
| 50 | 0.118 | 0.027 | 0.207 | 0.079 |
| 100 | 0.046 | 0.012 | 0.060 | 0.020 |

*4.3. Accuracy of Heating Rates*

In Table 5, the absolute RSME of heating rates, as determined via high spectral resolution simulations and weighted sum for the U.S. Standard Atmosphere, are shown for $n_{wvl} = 10, 30, 50$ and $100$. Low irradiance errors still result in low errors in the corresponding heating rates. Increasing the number of sampling nodes improves the accuracy of the heating rate calculations. In Figure 6, the heating rates for the U.S. Standard Atmosphere, calculated with 10, 30 and 50 sampling nodes and HSR-simulations, are depicted. The figure illustrates the improvement in accuracy with an increased number of sampling nodes.

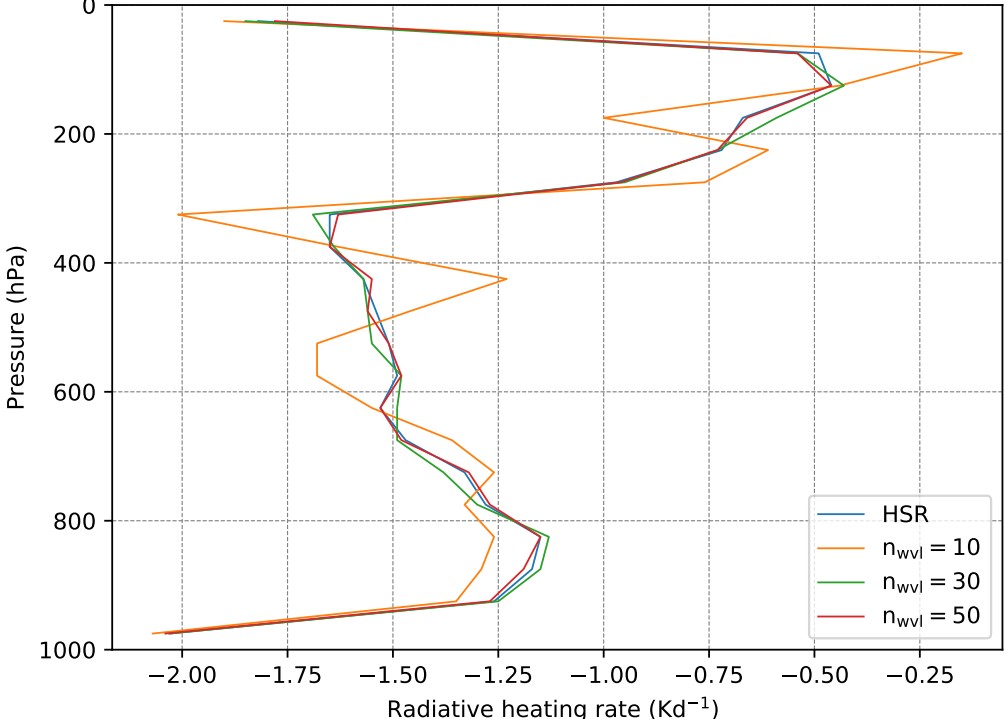

**Figure 6.** Heating rates of the U.S. Standard Atmosphere, calculated using different numbers of sampling nodes.

**Table 5.** Absolute RSME $\epsilon$ and maximum error $\epsilon_{\max}$ for heating rates of the U.S. Standard Atmosphere calculated with different numbers of sampling nodes. The unit for the values in the table is $[\mathrm{Kd}^{-1}]$.

| $n_{\mathbf{wvl}}$ | $\epsilon$ | $\epsilon_{\max}$ |
|---|---|---|
| 10 | 0.1817 | 0.3555 |
| 30 | 0.0323 | 0.0806 |
| 50 | 0.0178 | 0.0491 |
| 100 | 0.0049 | 0.0128 |

### 4.4. Cloudy Atmospheres

An important question posed in [6] is whether this method can be applied to atmospheric scenarios involving clouds.

We modeled such a cloud by uniformly adding a value of $\tau = 5$ to the spectral optical thickness at its intended height in the atmosphere. In Figure 7, the heating rates calculated with 10, 30 and 50 sampling nodes and HSR-simulations are depicted to add such a cloud at the 10th layer in U.S. Standard Atmosphere. This corresponds to a cloud between 500 hPa and 550 hPa.

We included this cloud layer in the 500 atmospheric profiles, whose irradiances were determined in Section 4.2. In Table 4, the resulting RSMEs for the same 500 atmospheres from [11] are shown to have an extra cloud. For each of those, a cloud was added to the 22nd layer (3.1 to 3.4 km) by again adding $\tau = 5$ to the spectral optical thickness.

Comparing these errors with the errors in the original atmospheres without a cloud in Table 4, the reduced lookup tables performed similarly, and slightly better for 10 sampling nodes. Since the errors were stable under the addition of a cloud, the validity of the reduced lookup tables for cloudy atmospheres is confirmed.

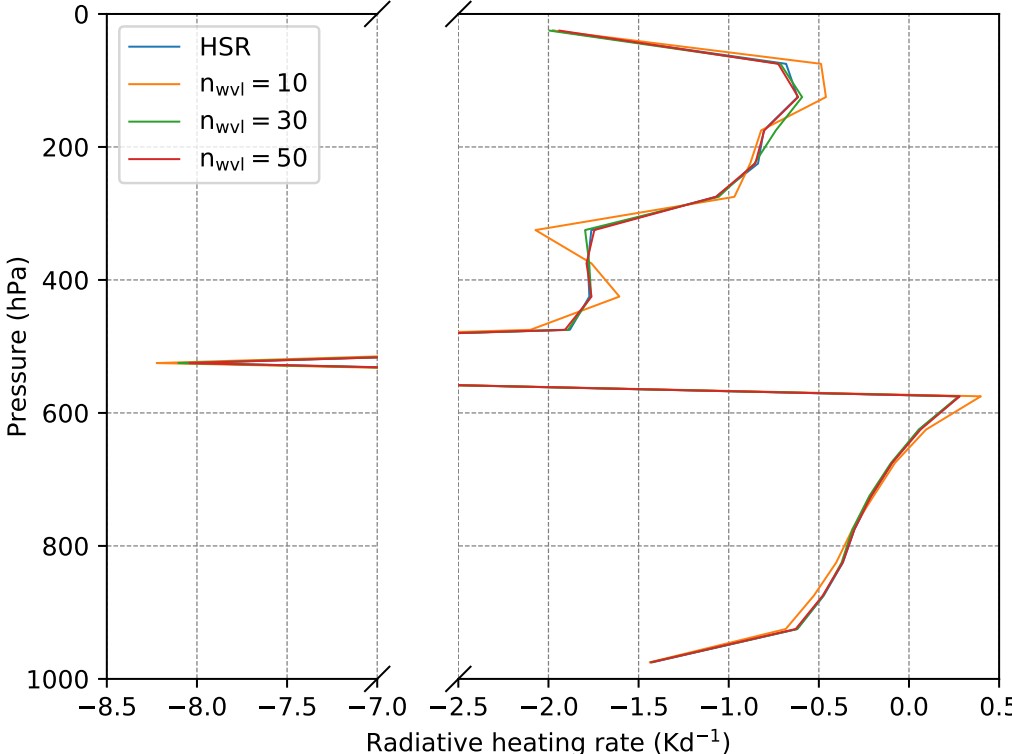

**Figure 7.** Heating rates of the U.S. Standard Atmosphere with a cloud calculated using different numbers of sampling nodes.

## 5. Conclusions

In this work, we used the simulated annealing algorithm to determine a small set of representative wavelengths to calculate integrated thermal irradiances and heating rates. Using a weighted sum approach, the computational cost of integral calculation could be decreased by several orders of magnitude. A cost-efficient lookup table approach was adopted to calculate the optical absorption thicknesses. Using a simple Schwarzschild radiative transfer model, we calculated irradiances at a high spectral resolution for various atmospheric scenarios, which could be used as training data for the simulated annealing algorithm. As the next step, for different numbers of sampling nodes, optimized lookup tables were produced, including representative wavelengths and corresponding weights. The integrated irradiance can be calculated from this table using a weighted sum of irradiances, which were calculated at the representative wavelengths. Through their application on a large test dataset, we found that ten representative wavelengths are sufficient to achieve an average RSME for irradiances below $1\,\mathrm{Wm^{-2}}$. With 100 wavelength nodes, an average RSME below $0.05\,\mathrm{Wm^{-2}}$ can be achieved. The method was verified for a large variety of atmospheric profiles taken from the ECMWF model. This performed equally well for atmospheres with or without clouds.

Throughout this work, one specific training dataset was used, which used typical atmospheric profiles from a numerical weather prediction (NWP) model. Alternatively, for radiative transfer in climate models, the training data could be extended by including higher variability in greenhouse gases to derive more precise lookup tables for these scenarios. This method could also be used to train specifically for heating rates, in order to achieve a higher accuracy in that area. The REPINT parameterization is available in the libRadtran radiative transfer package [15,16]. This can be used in combination with various radiative transfer solvers based on different methodologies, such as twostream or discrete ordinate, which consider scattering in 1D and 3D geometry.

In the future, we plan to extend the methodology to the solar spectral region. Here, scattering can no longer be neglected; therefore, another radiative transfer solver needs to be applied. The simulated annealing approach is expected to work equally well in the solar region, since it has already been used to generate the REPTRAN parameterization [7] for spectral bands in both solar and thermal spectral regions.

**Author Contributions:** Conceptualization, B.M., C.E. and M.d.M.; methodology, M.d.M., B.M. and C.E.; software, M.d.M. and C.E.; validation, M.d.M., B.M. and C.E.; formal analysis, M.d.M.; investigation, M.d.M.; resources, B.M. and C.E.; data curation, M.d.M.; writing—original draft preparation, M.d.M.; writing—review and editing, B.M. and C.E.; visualization, M.d.M.; supervision, B.M. and C.E.; project administration, B.M.; funding acquisition, B.M. All authors have read and agreed to the published version of the manuscript.

**Funding:** Funding was provided by subproject B4 of the Transregional Collaborative Research Center SFB/TRR 165 "Waves to Weather" www.wavestoweather.de (accessed on 27 October 2022) funded by the German Research Foundation (DFG).

**Institutional Review Board Statement:** Not applicable.

**Informed Consent Statement:** Not applicable.

**Data Availability Statement:** The parameterization used in this study is public and available at www.libradtran.org (accessed on 27 October 2022).

**Conflicts of Interest:** The authors declare no conflict of interest.

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
