# Peer review of "Optimized Wavelength Sampling for Thermal Radiative Transfer in Numerical Weather Prediction Models"

_atmosphere, doi:10.3390/atmos14020332_

Round 1

Reviewer 1 Report

The study titled "Optimized wavelength sampling for thermal radiative transfer in numerical weather prediction models" presents a method to simplify solving infrared irradiance and heating rate calculations by using weighted means at a relatively few number of spectral wavelengths with weights determined using an optimization technique known as annealing.  The authors present a straight forward, well written and explained study that is sufficient for publication in Atmosphere.  However, I suggest a few minor changes to clear up results and methodology to make the study stronger - most of which are related to the figures and tables.

Minor comments:

In all figures, please include grid lines to allow for better reading.

In all tables and in the text, please use consistent significant figures.  This allows for better alignment and clarity.

On Eq. 3 make sure to separate the two equations as it seems to be a single equation with how it is currently written.  It may be useful to break into two lines and label Eq. 3 and Eq. 4.

On line 91, please clarify which equation you mean by "The above equation".

On line 105, please define MT_CKD.

On line 106, please define ARTS.  I may have missed others, but please ensure that all acronyms are defined.

On line 142, there is a missing comma after "Hence".

On line 220, please expand "w.r.t.".

Figure 4 is likely better presented as a scatter plot.  It also may make sense to make the x-axis on a logarithm scale.  Finally, include labels (e.g., A, B, C, D) in the figures and explain what each subplot indicates in the caption.

Figure 5 - the x-axis label is listed as index.  Presumably this is profile number from the 5000 atmospheric profile subset.  Maybe find a better axis title for this.  It may also be beneficial to split this into atmosphere type (e.g., tropical, mid-latitude, etc).

As opposed to showing the cloudy vs clear-sky heating rate, it may be more useful to show the cloudy-sky version of Fig. 7.

Reviewer 2 Report

I found no errors in the article, but I feel I am not competent enough to make a critical substantive assessment of it. I leave it to the editor to decide whether or not to take my opinion into account.

Reviewer 3 Report

This manuscript presents a novel optimization-based approach to developing a computationally-efficient thermal infrared radiative transfer.  The authors note that correlated-k methods have been well-established to do just this, but that other approaches are gaining favor.  This work explores one such approach.  The manuscript, as presented, requires major revisions before it can be acceptable for publication.  There are some clear errors in what is presented, and, in addition to that, the authors make a number of choices in their experimental setup which are not justified. The errors, of course, need to be corrected, and also more explanation is needed. 

The result is that this Reviewer is left with the impression that the work needs to be substantially redone with a closer attention to detail.  For example, in the derivation section (Section 1) the mu and mu_0 variables are not explained, for example, and then mu is used elsewhere and has a different meaning.  While the implementation of simulated annealing to develop computationally-efficient thermal infrared radiative transfer is appealing, it must be implemented rigorously for it to be acceptable for publication in Atmosphere.

First, the errors:

1.     The spectrum shown in Figure 1 is clearly the result of an erroneous calculation.  The top-of-atmosphere irradiance around 15 um appears to exactly follow the manifold of Planck function emission, suggesting perhaps that a realistic stratospheric temperature profile (which is part of the US Standard Atmosphere) was not properly including in this calculation, or issues with the absorption cross-section of CO2, or the radiative transfer calculation.

2.     Page 4: “Upward irradiance at the surface as well as downward irradiance at the top-of-atmosphere are set to … 0”.  The upward irradiance at the surface can only be 0 if it has an emissivity of 1 and a temperature of 0 K.  This indicates another error in the radiative transfer formulation.

The authors will need to show realistic TOA infrared spectra and correct this error.  The revision response to authors will need to show irradiance spectra for all of the 42 atmospheres referenced to ensure that these first-order errors are not present in the revision.

Second, justifications:

1.      The use of HITRAN 2012 must be justified, given that there are several recent updates, including in 2016 and 2020.

2.     Page 4: The choice of doing the calculation in one propagation direction (mu=0.5) needs to be justified other than its computational expense savings (i.e., minimal impact on accuracy).

3.     Line 97: the averaging of values of adjacent levels to get layer quantities can introduce errors (https://agupubs.onlinelibrary.wiley.com/doi/full/10.1029/2003JD004457).  The authors need to show that these were minimized.

4.     Line 105: Which version of MT_CKD was used?  It has recently been updated substantially (https://github.com/AER-RC/MT_CKD)

5.     Why is this done in wavelengths?  The analysis, as presented, is quite confusing when done in wavelength space, especially since there is a focus on fluxes and heating rates.  It makes far more sense to present everything in frequency space, especially because the sampling of different parts of the thermal infrared spectrum are highly unevenly spaced in wavelength-space, but likely much more regularly-spaced in frequency space.

6.     Scattering is curiously missing, even though it can be significant for cloudy atmospheres (doi: https://doi.org/10.1175/1520-0469(1997)054<2799:MSPITI>2.0.CO;2 and many of the references therein).

Reviewer 4 Report

A gas radiation model which uses an optimized wavelength sampling method is developed to calculate radiative transfer in atmosphere efficiently. The parameters of the proposed gas radiation model is set based on 42 atmospheres and results show that the parameters are valid for a wider range of atmosphere profiles. The work is interesting and can be published if following comments can be well addressed.

1) To my opinion, the wavelength sampling method used in this work is somehow similar to a full spectrum gas radiation model named WEIGHTED-SUM-OFGRAY-GASES (WSGG) model (Modest, M. F.: “The weighted-sum-of-gray-gases model for arbitrary solution methods in radiative transfer,”ASME Journal of Heat Transfer, vol. 113, no. 3, pp. 650–656, 1991.). Can the authors explain what is the main differences between these two methods?

2) As described by Eqs. (8) and (11), the parameters used for the absorption cross sections of a general trace gas are different from those of water vapor. What is the reason? Does that mean the VMR change of a general trace gas is negligible for different atmosphere profiles?

3) The optimized parameters are shown to have good accuracy for the atmosphere profiles considered in this work. How about for other atmosphere profiles in other places of the world? Can the authors comment on this?

4) Eq. (3) should be revised and presented in two lines.

Round 2

Reviewer 4 Report

All my questions are well addressed.